# Systems analysis of miR-199a/b-5p and multiple miR-199a/b-5p targets during chondrogenesis

**Krutik Patel[1]\*[†], Matt Barter[2][†], Jamie Soul[2,3], Peter Clark[1], Carole Proctor[1], Ian Clark[4], David Young[2]\*, Daryl P Shanley[1]**

[1]Campus for Ageing and Vitality, Biosciences Institute, Newcastle University, Newcastle-upon-Tyne, United Kingdom; [2]Regenerative Medicine, Stem Cells, Transplantation, Biosciences Institute, Newcastle University, Newcastle upon Tyne, United Kingdom; [3]Computational Biology Facility, Faculty of Health and Life Sciences, University of Liverpool, Liverpool, United Kingdom; [4]School of Biological Sciences, University of East Anglia, Norwich, United Kingdom

**\*For correspondence:**
nkp68@ncl.ac.uk (KP);
david.young@newcastle.ac.uk (DY)

[†]These authors contributed equally to this work

**Competing interest:** The authors declare that no competing interests exist.

**Abstract** Changes in chondrocyte gene expression can contribute to the development of osteo-arthritis (OA), and so recognition of the regulative processes during chondrogenesis can lead to a better understanding of OA. microRNAs (miRNAs) are key regulators of gene expression in chondrocytes/OA, and we have used a combined experimental, bioinformatic, and systems biology approach to explore the multiple miRNA–mRNA interactions that regulate chondrogenesis. A longitudinal chondrogenesis bioinformatic analysis identified paralogues miR-199a-5p and miR-199b-5p as pro-chondrogenic regulators. Experimental work in human cells demonstrated alteration of miR-199a-5p or miR-199b-5p expression led to significant inverse modulation of key chondrogenic genes and extracellular matrix production. miR-199a/b-5p targets *FZD6*, *ITGA3* and *CAV1* were identified by inhibition experiments and verified as direct targets by luciferase assay. The experimental work was used to generate and parameterise a multi-miRNA 14-day chondrogenesis kinetic model to be used as a repository for the experimental work and as a resource for further investigation of this system. This is the first multi-miRNA model of a chondrogenesis-based system, and highlights the complex relationships between regulatory miRNAs, and their target mRNAs.

## eLife assessment

This study provides **valuable** insight into the role of miR-199a/b-5p in cartilage formation. The evidence supporting the significance of the identified miRNA and its target mRNA transcripts is **convincing**. This article will likely primarily benefit scientists focused on diseases related to this biological process, such as osteoarthritis. Furthermore, researchers with a broader interest in miRNAs may find the computational model to identify novel RNA–RNA interactions particularly helpful.

## Introduction

Differentiation of mesenchymal stem cells (MSCs) into chondrocytes occurs by the process of chondrogenesis (*Bosnakovski et al., 2006*). This is an important process during development as it is a pre-requisite for skeletogenesis. Chondrocytes perform the vital role of generating cartilage during embryogenesis, but also maintain cartilage throughout life, including at the ends of long bones in articulating joints (*Akkiraju and Nohe, 2015*). The master transcription factor responsible for

chondrogenesis is SOX9 and during this process chondrocytes secrete anabolic proteins such as type II collagen and aggrecan, encoded by *COL2A1* and *ACAN*, which constitute a significant functional portion of the cartilaginous extracellular matrix (ECM) (*Akkiraju and Nohe, 2015*; *Hoshi et al., 2017*). With increasing age and influenced by a mixture of factors such as (epi)genetics, obesity, and mechanical injury/stress, chondrocytes will increasingly express catabolic proteins such as matrix metalloproteinases which degrade the cartilage ECM (*Hoshi et al., 2017*). Ultimately the chronic loss of cartilage contributes to the extremely debilitating condition osteoarthritis (OA), which remains incurable with treatment options limited to pain relief medication and end-stage joint replacement surgery (*Grässel and Muschter, 2020*; *Loeser et al., 2012*).

miRNAs are short non-coding RNAs – roughly 19-22 nt long that negatively regulate target mRNAs (*Bartel, 2009*; *Sevignani et al., 2006*). Mammalian miRNA–mRNA interactions occur via complementary sequence specificity between a 7 and 8 nt stretch of the miRNA known as the seed sequence, and positions along the 3′UTR of the target mRNA (*Lai, 2002*; *Doench and Sharp, 2004*). miRNA–mRNA interactions are complex because a single mRNA can be targeted by many miRNAs, and a single miRNA can target many mRNAs. Our previous work identified significantly altered miRNAs during chondrogenesis and demonstrated the role of miR-140 in regulating chondrocyte gene expression (*Barter et al., 2015*). miR-140 has emerged as a vital regulator of chondrogenesis, cartilage, and OA, and have been hypothesized as effective drug targets due to their pro-chondrogenic regulation (*Miyaki et al., 2009*; *Katoh et al., 2021*; *Swingler et al., 2012*; *Karlsen et al., 2016*). Additional miRNAs, such as miR-455, have also been shown to be important regulators in maintaining healthy cartilage and protecting against OA development (*Hu et al., 2019*; *Ito et al., 2021*).

To identify further important miRNAs from this dataset, and to overcome the complexity of miRNA–mRNA interactions, we performed a combined bioinformatic, experimental and systems biology approach to better understand the relationship between miRNAs which may be important to chondrogenesis, and their predicted targets. We identified miR-199a-5p and miR-199b-5p as pro-chondrogenic miRNAs. Just as with miR-140-5p and miR-455, we anticipated miR-199a-5p and miR-199b-5p (miR-199a/b-5p) to target multiple mRNAs during chondrogenesis. To this end we used experimental, informatic, and literary data to build kinetic models to explain how miR-199a/b-5p regulated chondrogenesis. The models included three targets identified through RNAseq analysis (*FZD6*, *ITGA3*, and *CAV1*), and were used to make predictions to fill experimental gaps and predict novel interactions between miR-199a/b-5p and the chondrogenesis machinery.

## Results

### Analysis with *TimiRGeN* identified miR-199b-5p to be upregulated during chondrogenesis

Our previously generated 14-day chondrogenesis time-series dataset was analysed with the *TimiRGeN R/ Bioconductor* package – a novel tool we developed to analyse longitudinal miRNA–mRNA expression datasets (*Barter et al., 2015*; *Patel et al., 2021*). Prior to using *TimiRGeN*, the transcriptomic data underwent timepoint-based differential expression analysis using the day 0 timepoint (MSCs) as the control group for chondrogenesis samples measured at days 1, 3, 6, 10, and 14, and thus created differential expression data over five timepoints. *TimiRGeN* identified signalling pathways of interest over the 14-day time-series data (*Supplementary file 1a*). Eight signalling pathways were found enriched in at least three of the five timepoints (*Figure 1*). The *TimiRGeN* pipeline was then used to predict miRNA–mRNA interactions that may regulate each of the eight pathways. miRNA–mRNA interactions were kept if the miRNA and mRNA involved in the interaction has a Pearson correlation <−0.75 across the time series and if the interaction was predicted in at least two of the following three databases: TargetScan, miRDB, or miRTarBase (*Agarwal et al., 2015*; *Huang et al., 2020*; *Chen and Wang, 2020*). To identify which miRNAs involved in the miRNA–mRNA interactions were positively changing in each of the eight signalling pathways, we scaled the log2FC values from *limma* (*Ritchie et al., 2015*). By using scaled log2FC values, we could highlight magnitude of change, rather than total change (*Figure 1B*).

miR-140-5p was the most positively changing miRNA in the following seven pathways: adipogenesis, clear cell renal cell carcinoma pathways (CCRCCP), epidermal growth factor/epidermal growth

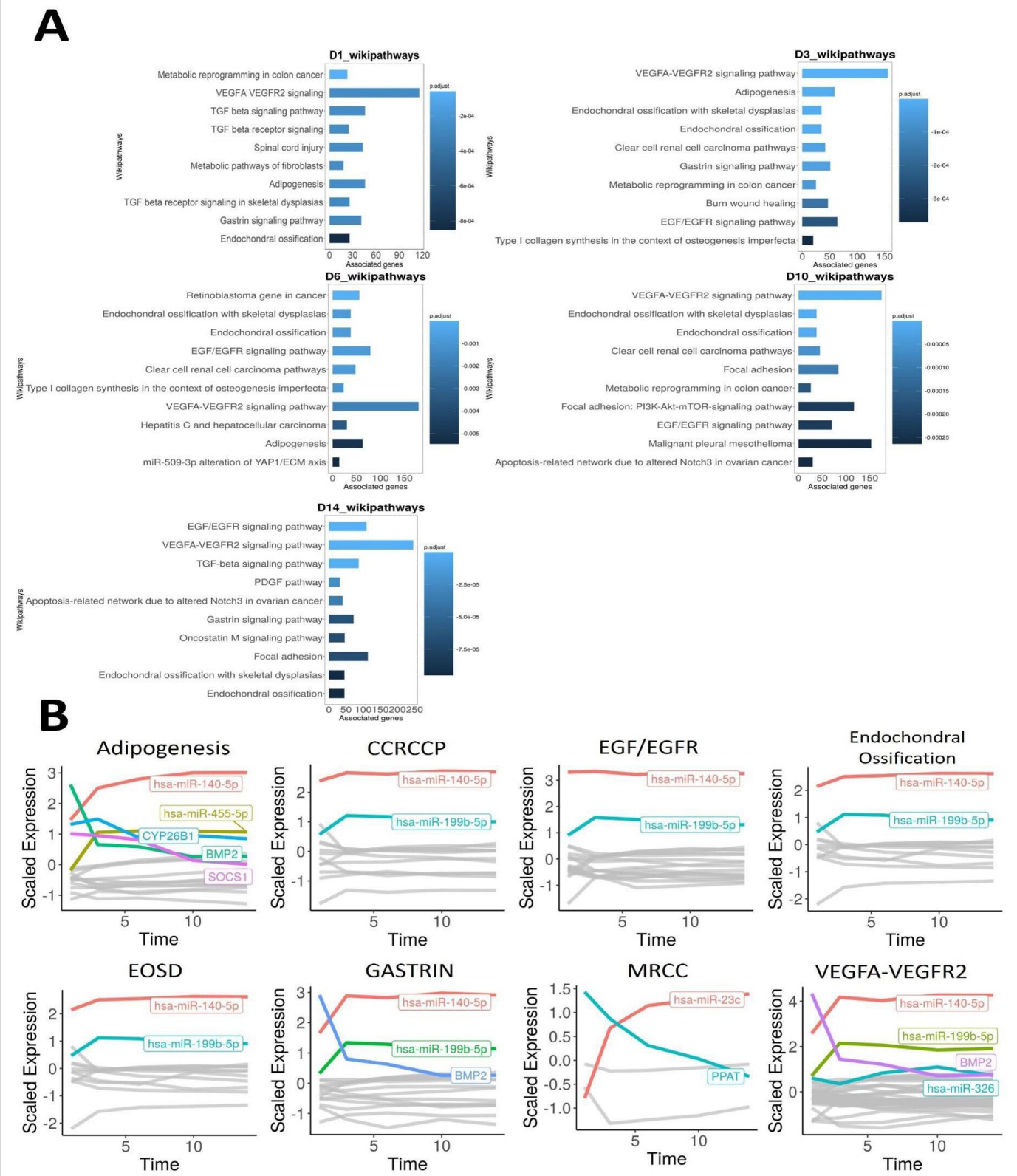

**Figure 1.** Time-course bioinformatic identification of miR-199 as a key regulator of chondrogenic gene expression. (**A**) Overrepresentation analysis (ORA) of the differentially expressed miRNAs and mRNAs at each timepoint contrasted to D0. For each significantly enriched pathway identified, the number of associated genes found from the pathway is shown on the x-axis. (**B**) Line plots displaying scaled log2FC values over the 14-day time course for the indicated pathways. Acronyms are defined the text. MRCC = metabolic reprogramming in colon cancer. Here, individual genes found in the

*Figure 1 continued on next page*

Figure 1 continued

filtered miRNA–mRNA interactions for each pathway are plotted along a time course. Only genes (miRNAs or mRNAs) that have a scaled log2FC value of at least 1 at any point of the line plot are highlighted and labelled.

factor receptor (EGF/EGFR), endochondral ossification, endochondral ossification with skeletal dysplasia (EOSD), gastrin signalling pathway, and vascular endothelial growth factor-A/vascular endothelial growth factor receptor 2 signalling pathway (VEGFA-VEGFR2). miR-199b-5p was the second most positively changing microRNA in the following six pathways: CCRCCP, EGF/EGFR, endochondral ossification, EOSD, gastrin signalling pathway, and VEGFA-VEGFR2. Other miRNA/genes such as hsa-miR-455-5p and *BMP2* were also of interest, but we focused on hsa-miR-199b-5p and its paralogue miR-199a-5p since these are comparatively under-researched miRNAs within the context of chondrogenesis (*Barter et al., 2015*).

## Expression of chondrogenic biomarkers and glycosaminoglycan levels change over time after altering miR-199a-5p or miR-199b-5p expression

To identify if miR-199a-5p and miR-199b-5p regulate chondrogenesis, we performed MSC chondrogenic differentiation and measured the consequences of miR-199 overexpression and inhibition (*Figure 2—figure supplement 1*) on chondrogenic biomarkers *ACAN*, *COL2A1*, and *SOX9* and glycosaminoglycan (GAG) levels. Significant upregulation was seen in *ACAN*, *COL2A1*, and GAG (DMB) levels following miR-199a-5p overexpression (*Figure 2A*). In contrast, when miR-199a-5p expression was inhibited *ACAN*, *COL2A1*, and *SOX9* were significantly downregulated at multiple timepoints (*Figure 2B*). GAG levels were also significantly decreased, by ~40%, at day 7. When miR-199b-5p was inhibited, similar significant downregulation of *ACAN*, *COL2A1*, and GAG levels occurred (*Figure 2C*). Inhibition of miR-199a-5p and -199b-5p together caused a more consistent reduction of *ACAN*, *COL2A1*, and *SOX9* expression.

## *FZD6*, *ITGA3*, and *CAV1* are the most significantly upregulated miR-199a/b-5p targets during MSC chondrogenesis

Multiple mRNA targets of miR-199a/b-5p are likely regulated during chondrogenesis. To elucidate which genes are most affected by miR-199 inhibition, we performed an RNAseq experiment to identify candidate targets during the early phase of MSC chondrogenesis comparing MSC samples which were undifferentiated (day 0) and MSC samples which were in the early stages of chondrogenesis (day 1). Chondrogenic markers were downregulated by miR-199 inhibition in the early stages of chondrogenesis (*Figure 2*), so we reasoned that identifying mRNAs which are positively enriched during the first few days of chondrogenesis may identify the most important mRNA targets of miR-199a/b-5p. We chose to inhibit miR-199a-5p or miR-199b-5p since supraphysiological overexpression of miRNA mimics can lead to spurious findings (*Jin et al., 2015*). Initial comparison between day 1 differentiated and day 0 undifferentiated control samples identified 4391 upregulated and 4168 downregulated significantly differentially expressed genes (DEGs) (<0.05 adjusted p-value). Positively changing genes included *COL2A1* (log2fc = 11.6), *ACAN* (log2fc = 9.18), and *SOX9* (log2fc = 3.37). Gene Ontology (GO) term analysis on the upregulated genes confirmed that the cells were differentiating towards chondrocytes with terms such as skeletal system development, ECM organisation, and regulation of cartilage development significantly (adjusted p-values<0.05) enriched (*Supplementary file 1b*).

To identify miR-199-regulated genes, undifferentiated control MSC samples were contrasted against undifferentiated MSCs with either miR-199a-5p or miR-199b-5p inhibition, which respectively resulted in 87 and 46 significantly DEGs (adjusted p-values<0.05) (*Supplementary file 1*). Similar comparisons at day 1 of chondrogenesis revealed inhibition of miR-199a-5p or miR-199b-5p respectively resulted in 674 and 817 DEGs. Here, 25 and 341 genes intersected between day 0 and day 1 chondrogenesis inhibition experiments, indicating that both microRNAs may share functional repertoire of targets (chi-square observed vs. expected day 0 = 0.003477, day 1 = 0.000624). *COL2A1*, *ACAN*, and *SOX9* were significantly lower in expression in day 1 samples after miR-199 inhibition, validating the negative impact on chondrogenesis. GO term analysis was conducted to identify biological processes linked to miR-199a/b-5p inhibition (*Figure 3A*). Interestingly several terms associated with

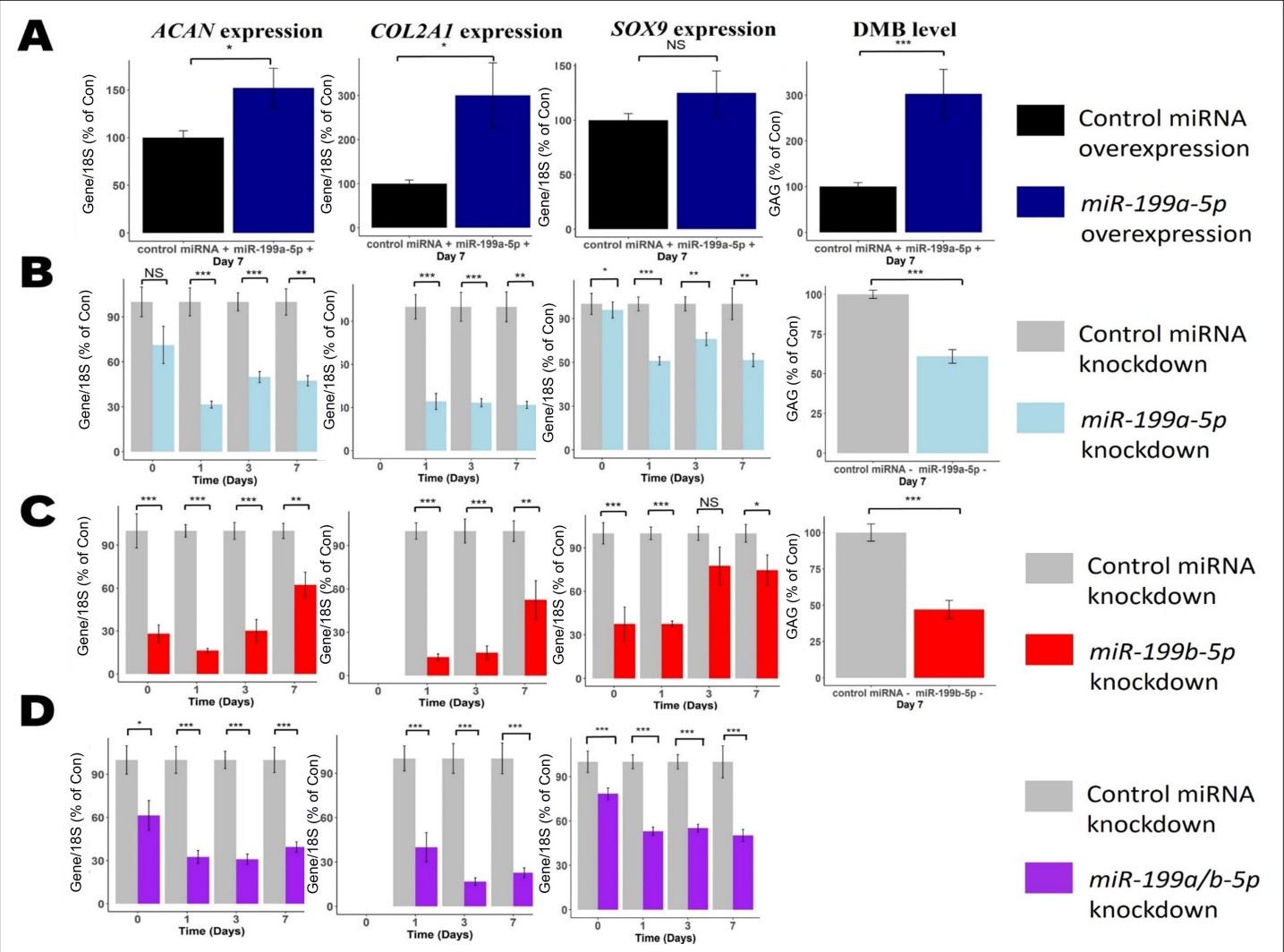

**Figure 2.** Modulation of miR-199 affects chondrogenesis gene expression and extracellular matrix (ECM) production. Mesenchymal stem cells (MSCs) were transfected for 24 hr with miR-199a-5p, miR-199b-5p, or non-targeting miRNA control mimics or inhibitors prior to the induction of chondrogenesis. (**A**) Overexpression of miR-199a-5p. (**B–D**) Inhibition of (**B**) miR-199a-5p (**C**), miR-199b-5p, or (**D**) miR-199a-5p and -199b-5p. (**A–D**) At days 0, 1, 3, and 7 after initiation of chondrogenesis, RNA was extracted and measurements of *ACAN*, *COL2A1*, and *SOX9* gene expression were taken. qPCR results for day 0 were undetectable for *COL2A1*. Gene expression was normalised to 18S. Values are the mean ± SEM of data pooled from 3 to 4 separate MSC donors (N=3-4), with 4–6 biological replicates per donor (n=4-6). Presented as % of non-targeting control levels. The p-values calculated by paired two-tailed Student's *t*-test, NS = not significant, *<0.05, **<0.01, ***<0.001.

The online version of this article includes the following figure supplement(s) for figure 2:

**Figure supplement 1.** Mesenchymal stem cells (MSCs) were transfected for 48 hr with (**A**) miR-199a-5p mimic (mi) or (**B**) miR-199a-5p/199b-5p hairpin (hp) inhibitor, or non-targeting controls (Con2).

chondrogenesis were suppressed/less activated during day 1 analyses, including ECM constituents, skeletal system morphogenesis, ECM, and collagen containing ECM.

The significantly differentially expressed genes were also analysed using the miRNAtap R package to predict miR-199a-5p and miR-199b-5p targets (***Figure 3B***). Twenty-one potential mRNA targets were predicted to be targeted by both miRNAs, and in alphabetical order they were *ABHD17C*, *ATP13A2*, *CAV1*, *CTSL*, *DDR1*, *FZD6*, *GIT1*, *HIF1A*, *HK2*, *HSPA5*, *ITGA3*, *M6PR*, *MYH9*, *NECTIN2*, *NINL*, *PDE4D*, *PLXND1*, *PXN*, *SLC35A3*, *SLC35D1*, *VPS26A*. The expression patterns of these 21 miR-199 target genes were also explored in our microarray study (***Supplementary file 1c***; ***Barter et al., 2015***). SkeletalVis was used to contrast the behaviour of the 21 genes in other MSC-derived chondrogenesis studies (***Supplementary file 1d***; ***Soul et al., 2019***; ***Huang et al., 2010***; ***Huynh et al.,***

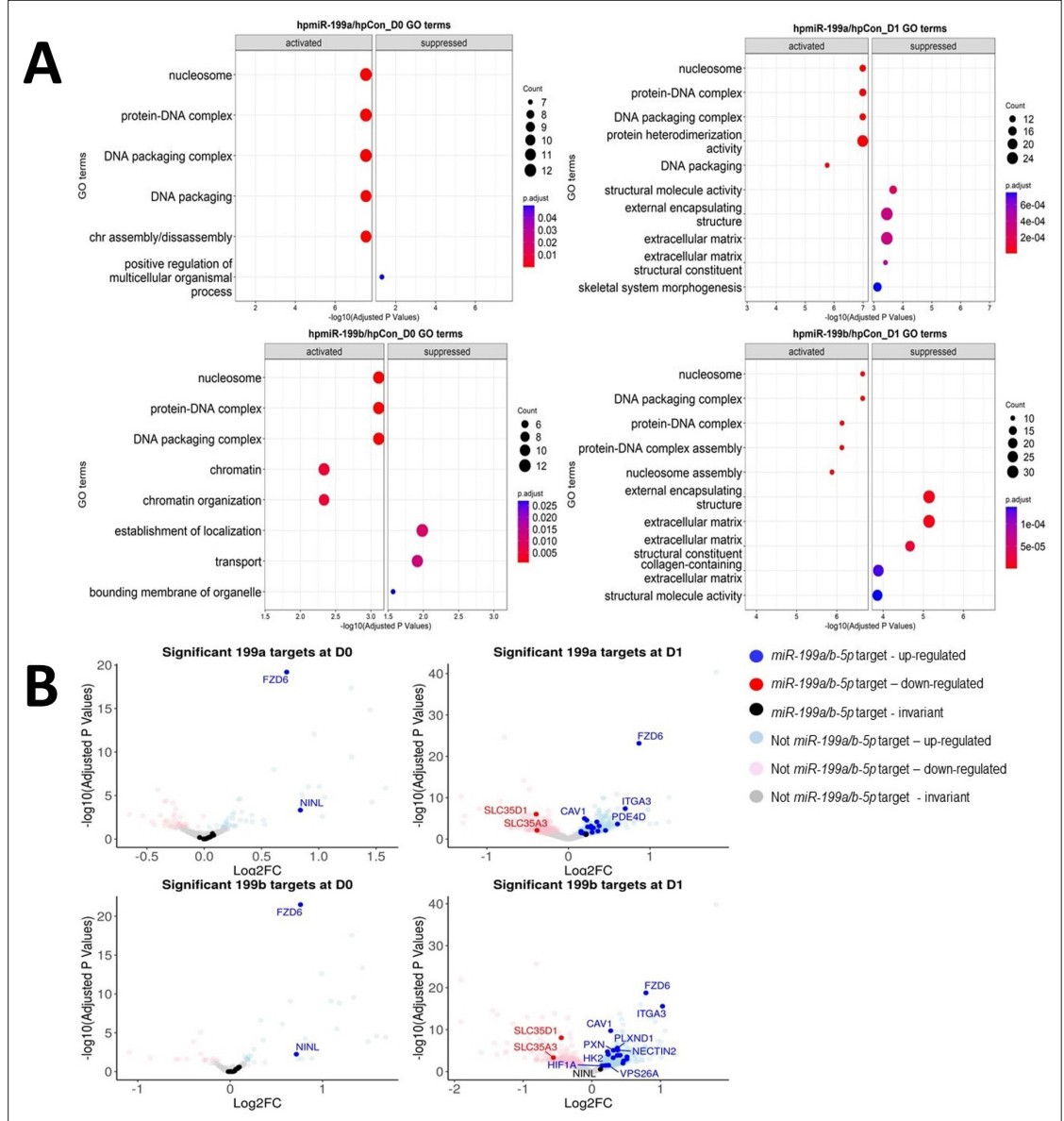

**Figure 3.** Identification of miR-199 targets during early chondrogenesis. Results from RNAseq analysis of control miRNA, miR-199a-5p, and miR-199b-5p inhibition experiments. (**A**) Gene Ontology (GO) analysis for the significantly differentially expressed genes found from miR-199a-5p or miR-199b-5p inhibition at day 0 and day 1 of chondrogenesis. Up to five activated and five suppressed pathways are displayed for each contrast. All GO terms shown have an adjusted p-value of <0.05. Count size represents the genes found in a pathway and this determined the size of the circles. (**B**) Volcano plots to display gene expression changes following inhibition of miR-199a or miR-199b at day 0 (D0) or day 1 (D1). The *miRNAtap*-selected 21 miR-199a/b-5p targets are identified (and labelled, space permitting) in red or blue if up- or downregulated The cut-off for significance was an adjusted (BH) p-value of <0.05. miR-199a/b-5p targets were upregulated by miR-199a/b-5p inhibition.

The online version of this article includes the following figure supplement(s) for figure 3:

**Figure supplement 1.** The 21 genes found through our analysis are displayed using several metrics: adjusted p-values are denoted by the colour of the shapes, the shapes reflect the time the sample was taken, and the size of the shapes represents the mean count of the transcripts abundance.

*2019*). The tables showed many of genes found through our bioinformatic analysis to be significantly downregulated during chondrogenesis in multiple studies, and this included *CAV1, FZD6, ITGA3,* and *MYH9*. We decided to further explore if *FZD6, ITGA3,* and *CAV1* were true targets of miR-199a/b-5p because in terms of adjusted p-values these three showed the most significant change (*Figure 3B, Figure 3—figure supplement 1*).

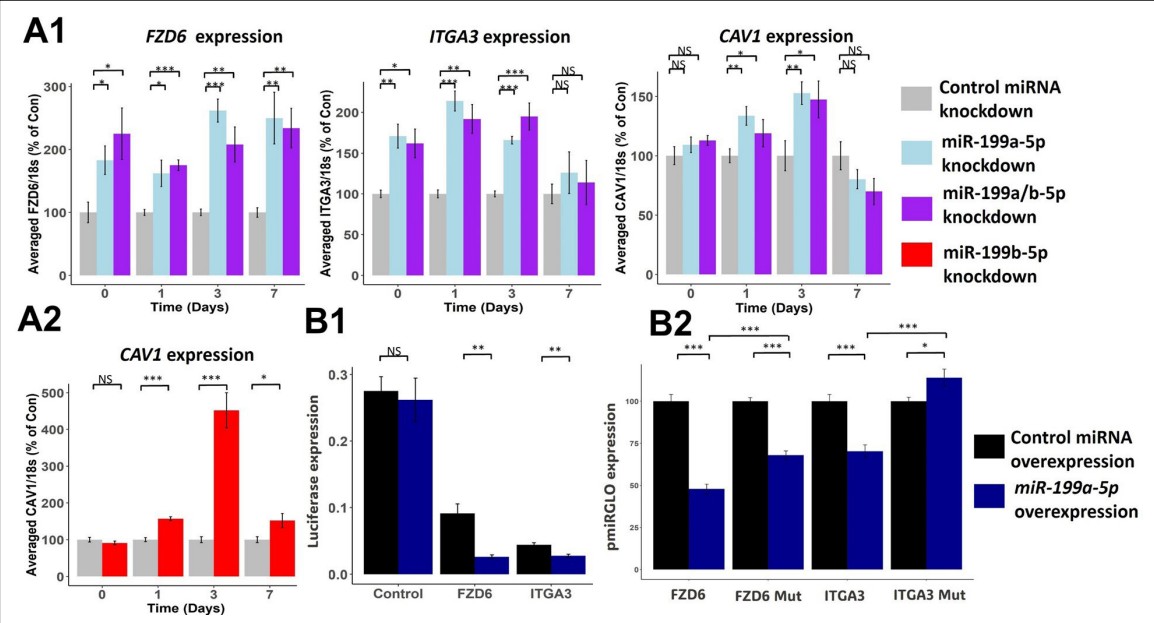

**Figure 4.** Effect of miR-199a/b-5p inhibition on putative miR-199 targets. (**A1–2**) Mesenchymal stem cells (MSCs) were transfected for 24 hr with miR-199a-5p, miR-199a/b-5p, miR-199b-5p, or non-targeting miRNA control inhibitors prior to the induction of chondrogenesis. At days 0, 1, 3, and 7 after the initiation of chondrogenesis, RNA was extracted and (**A1**) FZD6, ITGA3, and CAV1 expression was measured after miR-199a-5p and miR-199a/b-5p inhibition, or (**A2**) CAV1 levels were also measured after miR-199b-5p inhibition. Gene expression was normalised to 18S. Presented as % of non-targeting control levels. (**B1–2**) Luciferase expression in SW1353 cells following co-transfection of miR-199a-5p or non-targeting control mimic and miR-199a-5p target 3'UTR reporter constructs for 24 hr. (**B1**) FZD6 and ITGA3 3'UTR-regulated expression normalised to renilla luciferase. (**B2**) Wildtype and mutant FZD6 and ITGA3 3'UTR-regulated expression normalised to renilla and presented as percentage of non-targeting control levels. Values shown are the mean ± SEM of data pooled from (**A**) three separate MSC donors (N=3), with 4–6 biological replicates per donor (n=4-6), or (**B**) three independent experiments (n=3). p-Values were calculated using paired two-tailed Student's *t*-test, NS = not significant, *<0.05, **<0.01, ***<0.001.

To confirm the effects miR-199a-5p and miR-199b-5p had on the targets identified through RNAseq, we performed a series of inhibition experiments. Since the miRNAs shared the same seed site sequence (nucleotides 2–8; 5'-CCAGUGUU-3'), we tested if inhibition of one miRNA would lead to similar or different effects to inhibition of both miRNAs (*Figure 2—figure supplement 1*). For this, we picked to suppress expression of miR-199a-5p as it was the more highly expressed of the two paralogues and contrasted it to inhibition of miR-199a/b-5p. We tested the effects of the inhibition by measuring the expression of the three most significantly enriched predicted miR-199a/b-5p targets from our bioinformatic analysis – *FZD6*, *ITGA3*, and *CAV1* (*Figure 4A1*). We saw significant upregulation of *FZD6*, *ITGA3*, and *CAV1* after miR-199a-5p inhibition and miR-199a/b-5p inhibition. There was no consistent improvement of the combination of miR-199a-5p and miR-199b-5p over miR-199a-5p alone. We also tested the third most significant predicted target of miR-199a/b-5p – CAV1, and how its expression changed after miR-199b-5p inhibition over the same time course, and we saw CAV1 levels were significantly increased post miR-199b-5p inhibition (*Lino Cardenas et al., 2013*; *Figure 4A2*).

Unlike *FZD6* and *ITGA3*, *CAV1* has previously been established to be a direct target of miR-199a-5p (*Lino Cardenas et al., 2013*). To identify if *FZD6* and *ITGA3* were also direct miRNA targets, we cloned the 3'UTR regions of *FZD6* and *ITGA3* directly downstream of the firefly luciferase gene and demonstrated reduced expression compared to empty control vector, suggesting that these contain potentially repressive elements (*Figure 4B1*). Introduction of an excess of miR-199a-5p into the cells further repressed expression, suggesting that both FZD6 and ITGA3 3'UTRs are direct miR-199a-5p targets. This was confirmed by mutation of the predicted miR-199-5p seed sequence within the 3'UTR of each gene which reduced the extent of inhibition of luciferase levels caused by the miRNA (*Figure 4B2*).

## Kinetic modelling creates an in silico demonstration of how miR-199a-5p/miR-199b-5p regulates chondrogenesis

We attempted to capture the complexity of how miR-199a/b-5p regulated chondrogenesis using an in silico model (*Figure 5A*). Only using the experimental data presented in this article, and our previous microarray work, we developed a model to demonstrate the relationships between chondrogenic biomarkers and the targets of miR-199a/b-5p we identified through RNAseq and subsequent inhibition experiments. Finally, we used events to simulate inhibition of miR-199a-5p or miR-199b-5p (*Figure 5B*). The experiments shown in *Figures 2A, C and 4A* were used to parameterise the model, and since the experiments were performed in a staggered manner, we can use the model to make predictions to fill experimental gaps. Using this model, we predicted the dynamics of the chondrogenesis biomarkers and *CAV1* after miR-199a-5p inhibition and the dynamics of *FZD6* and *ITGA3* after miR-199b-5p inhibition. Most objects within the model were based on experimental data, and the differences between the experimental data and simulated data are calculated by mean squared error (MSE). In the initial model, 15/18 modelled objects with experimental data had an MSE of lower than 3, indicating that most of the experimental data was captured by the model, and the average MSE for the model was 15.96.

To enhance our initial model, we added further detail to increase biological relevance. This included the addition of transforming growth factor beta (TGFB) as a trigger for chondrogenesis initiation – just as we had during our chondrogenesis MSC experiments. We found strong mechanistic links between TGFB signalling and *CAV1* expression via SRC kinase. Based on our previous work, we also included miR-140-5p as it has been proven as a vital regulator of chondrogenesis, and with our kinetic model we predicted how miR-199a-5p or miR-199b-5p inhibition indirectly affected miR-140-5p. Further to this, we identified several flaws in our initial model, which we attempted to rectify using the enhanced chondrogenesis model. Firstly, by simulating the miR-199a-5p and miR-199b-5p inhibition to last until day 7, we saw that *SOX9* mRNA, *COL2A1* mRNA, and *ACAN* mRNA dynamics between days 1 and 7 of miR-199b-5p inhibition were flat. Furthermore, our CAV1 dynamics were also flat during miR-199b-5p inhibition. *Figure 2C* shows, from miR-199b-5p inhibition, the chondrogenesis biomarkers had a sharp decrease followed by a steady rise until day 7, and to match the effect of the miR-199b-5p inhibition experiments, we simulated the inhibition to last until day 4.5 instead of day 7. Doing so increased the similarities between our experimental data and simulations. Secondly, we wanted to include other miR-199a/b-5p targets which were alluded to in our RNAseq experiment, but not further explored, such as *MYH9* and *PDE4D*. We added a blackbox named 'OtherTargets' to represent other targets which miR-199a/b-5p regulate during chondrogenesis. Also, to delay the decrease in *ACAN*, *COL2A1*, and GAG levels after miR-199a/b-5p inhibition, we slowed down the interactions between the miR-199a/b-5p targets and *SOX9* by including SOX9 protein and SOX9 phospho-protein as objects in the model (*Figure 5C and D*). In the enhanced model, 14/18 (77.7%) of the objects with experimental data had an MSE of >3, indicating the model – even with additional data from the literature – still captured much of the experimental data. Also, the average MSE was 12.08, indicating an improvement over the initial model.

From the enhanced model, we saw improved dynamics for several model objects, for example, GAG, *CAV1*, *SOX9*, *ACAN*, *COL2A1*. Our MSE values which were used to quantify how similar our models' simulations were in contrast to our experimental data. Generally, our MSE values were better in our enhanced model for the chondrogenesis biomarkers and GAG levels, though our MSE values were better in our initial model for the miR-199a/b-5p targets. Overall, we saw the dynamics improved in the enhanced model, and in addition, we could make predictions on how miR-140-5p levels would indirectly be influenced from knock-down of miR-199a-5p or miR-199b-5p and such a multi-miRNA model has not yet been created before. Further pathways such as the Wnt signalling pathway may also play an important role in this system but could not be explored via modelling with our current level of data. We developed a hypothesis-rich GRN to display further pathways which were not in the scope of this project (*Figure 5—figure supplement 1*).

## Discussion

Our initial work in this area used a combined experimental and bioinformatics approach to identify and study the roles of miR-140 and miR-455 which were highly important to chondrogenesis (*Barter et al.,*

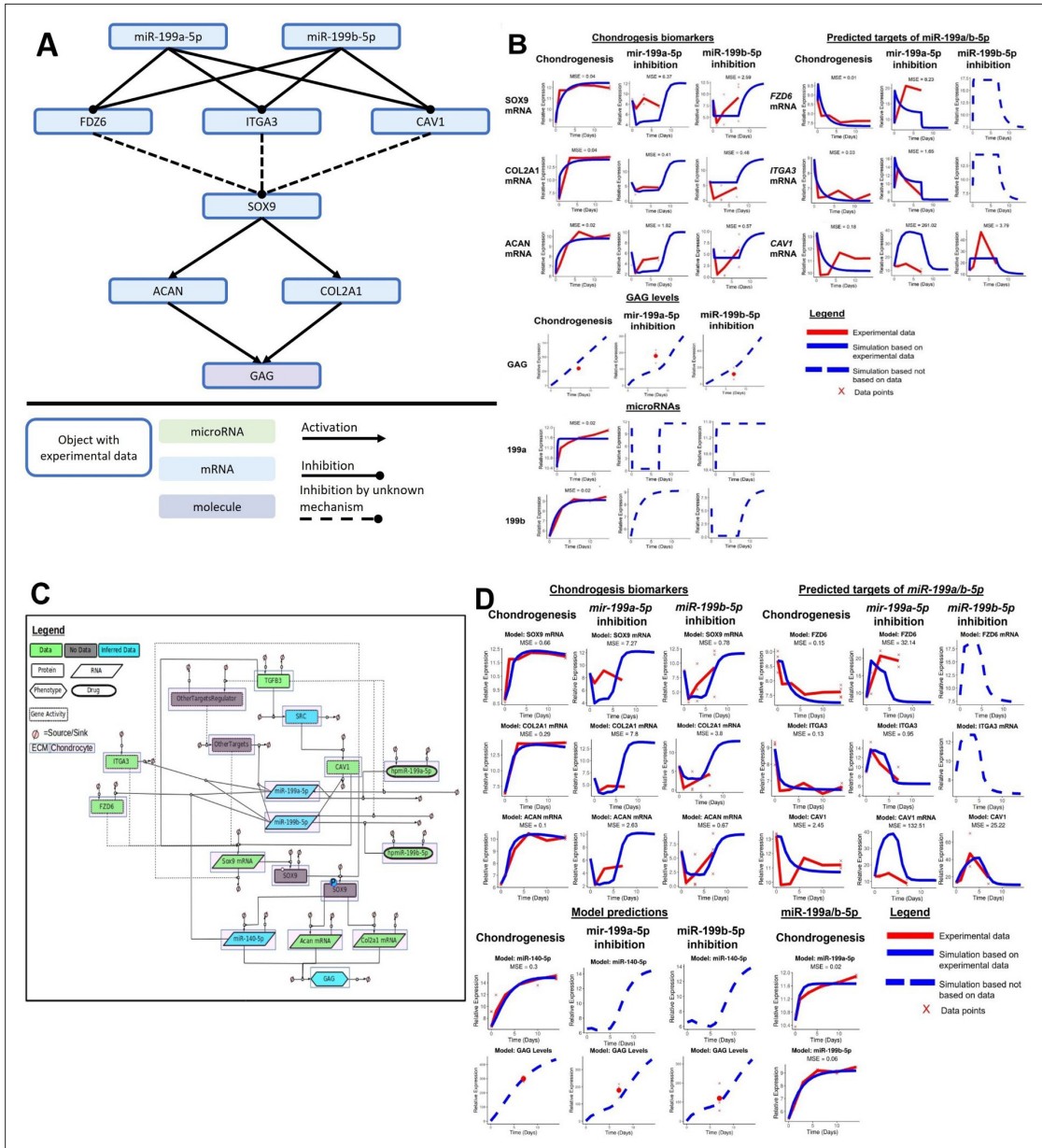

**Figure 5.** Initial kinetic modelling of miR-199a/b-5p regulation of chondrogenesis. (**A**) Schematic of how miR-199a/b-5p modulation effects the predicted miR-199a/b-5p *FZD6*, *ITGA3*, and *CAV1*, and the chondrogenic biomarkers *SOX9*, *COL2A1*, *ACAN*, and GAG. (**B**) Simulations (blue lines) from the kinetic modelling were contrasted against the experimental data – if available (red line) and a mean squared error (MSE) score is provided in these cases. Alternatively, if no experimental data was available, a dashed blue line displays the predicted behaviour of the gene. If multiple measurements were available, they have been displayed using red crosses. (**C**) A more detailed model displaying how miR-199a/b-5p regulates chondrogenesis via *FZD6*, *ITGA3*, and *CAV1* mRNAs, in GRN form. Here, information from the literature was added and miR-140-5p was added to the model. The GRN shown here is a minimalistic version of *Figure 5—figure supplement 1*. This was used to inform the topology of a kinetic model which aimed to explain how miR-199a-5p and miR-199b-5p act as pro-chondrogenic regulators by downregulating activity of *FZD6*, *ITGA3*, and *CAV1* mRNAs. This GRN contained 18 species including two proteins (TGFB3, SOX9), one phospho-protein (phospho-SOX9), three mRNAs (*SOX9*, *ACAN*, *COL2A1*), three miRNAs (miR-140-5p, miR-199a-5p, miR-199b-5p), two drugs (hpmiR-199a-5p, hpmiR-199b-5p), six protein activity (SRC, CAV1, FZD6, ITGA3, OtherTargets, OtherTargetsRegulator), and one phenotype (GAG). Each species has a sink and a source. Species are also shaped based on their properties: Proteins are rectangles, RNAs are rhombus, phenotypes are hexagons, drugs are oval, and gene activity are rectangles with dotted lines. Species are also highlighted with a white box if they are found in the extracellular matrix (ECM) or pink if they are found within a chondrocyte. Edges between species are solid if there is literature/data supporting an interaction or dotted if there the interaction is hypothetical. Species are also colour coded: green if there is associated data, blue if there is some data and the rest has been inferred based on literature, or grey if there is no data associated with the species. (**D**) Simulations from modelling the more detailed miR-199a/b-5p chondrogenesis model. Notations follow (**B**).

*Figure 5 continued on next page*

*Figure 5 continued*

The online version of this article includes the following figure supplement(s) for figure 5:

**Figure supplement 1.** GRN containing the broader scope of the biological system modelled.

---

*2015*; *Swingler et al., 2012*). This current work extends this, to identify other miRNAs of regulatory importance during chondrogenesis utilising a recently developed tool (*TimiRGeN* R/Bioconductor package) (*Patel et al., 2021*). Here, we combined experimental, bioinformatic, and systems biology approaches to identify and study the role of miR-199a/b-5p during chondrogenesis. The *MIR199* family were identified as pro-chondrogenic in mouse mesenchymal C3H10T1/2 cells, reportedly by targeting *SMAD1* (miR-199a-3p) and *JAG1* (miR-199a/b-5p), respectively (*Zhang et al., 2020*; *Lin et al., 2009*). Having previously shown the substantial upregulation of miR-199a/b-5p during chondrogenesis, we now show through both loss- and gain-of-function experiments that *miR-199a/b-5p* also promotes *SOX9, COL2A1,* and *ACAN* expression in human MSC chondrogenesis, thus enhancing cartilage formation (*Barter et al., 2015*). RNAseq analysis identified key targets of *miR-199a/b-5p* including *FZD6, ITGA3,* and *CAV1*, which were experimentally validated and subsequently incorporated into an in silico kinetic model.

miR-199a and miR-199b are vertebrate-specific miRNAs which exhibit an expression pattern associated with mesenchymal tissues and the skeleton (*Desvignes et al., 2014*). Antisense transcript *Dmn3os* encodes miR-199a and miR214 and its deletion in mice causes skeletal defects including short stature and cranial deformity (*Watanabe et al., 2008*). Similar phenotypic consequences caused by the loss of *DNM3OS* and therefore *MIR199A* and *MIR214* have also been reported in humans (*Lefroy et al., 2018*). *miR-214* has since been reported to negatively impact on chondrocyte differentiation, through targeting *ATF4* (*Roberto et al., 2018*). Thus, our demonstration of the pro-chondrogenic nature of miR-199a-5p and miR-199b-5p in human MSCs, in addition to the previous demonstration in mouse C3H10T1/2 cells, further supports the role of these miRNAs in the chondrogenesis and formation of the skeleton.

We show for the first time functional evidence for the miR-199b-5p-*CAV1* interaction to occur in human MSCs. Several mechanisms reported in the literature may also support how miR-199a/b--5p-*CAV1* could regulate chondrogenesis, such as TGFB triggering phosphorylation of CAV1 via SRC-kinase (*Lino Cardenas et al., 2013*). In contrast to *CAV1*, *FZD6* and *ITGA3* have been less well studied as miR-199a/b-5p targets though, miR-199a-5p-*FZD6* has been predicted previously (*Lin et al., 2009*; *Kim et al., 2015*). Our luciferase assays validate for the first time that *FZD6* is a target of miR-199a-5p. Previous work by our group has validated *FZD6* as a target of miR-140-5p, so it is likely a highly important gene in chondrogenesis. *ITGA3* meanwhile has been found to be a target of miR-199a-5p in neck and head cancer cells, and our results confirm this interaction in humans MSCs (*Tian et al., 2020*).

The presented model topology (*Figure 5A and C*) was based on our experimental work. Our initial model (*Figure 5A*) was reworked to better match the experimental patterns seen and to include additional genes which we can make prediction from. The enhanced chondrogenesis model (*Figure 5C*) was initiated with TGFB3, which was used to initiate chondrogenesis in our experiments and was therefore used as the proxy for chondrogenic initiation. As such, TGFB3 promoted miR-199a/b-5p levels to match our microarray data, though based on our previous data TGFB may not directly regulate miR-199a/b-5p levels (*Barter et al., 2015*). TGFB3 also induced CAV1 via SRC kinase and induced *SOX9* via the SMAD2-SMAD3 pathway, which then stimulated SOX9 protein production (SMAD2/3 were not included in the models) (*Coricor and Serra, 2016*; *Peng et al., 2008*; *Mishra et al., 2007*). However, our microarray data showed *CAV1* gene expression decreased during early chondrogenesis, but then increased again – perhaps indicating CAV1 has a smaller negative regulative effect on chondrogenesis, or only effects early chondrogenesis. Gene expression of *FZD6, ITGA3, CAV1* and the OtherTargets blackbox all had inverse relationships with miR-199a-5p and miR-199b-5p, and *FZD6* was also negatively regulated by miR-140-5p. *SOX9* mRNA, SOX9, and P-SOX9 were all treated as separate objects in this model. P-SOX9 promoted *COL2A1, ACAN,* and miR-140-5p expression. *ACAN* contributed directly, and to the greatest extent, to GAG levels (since most cartilage GAGs are post-translationally added to Aggrecan, the protein product of *ACAN*) which was used as the phenotypic level output for the model and a proxy for chondrogenesis progression (*Barter et al., 2015*; *Roughley and Mort, 2014*; *Hardingham and Fosang, 1992*;

*Huang et al., 2000*). Precise mechanisms of how FZD6 and ITGA3 regulated chondrogenesis are unclear, with potentially implicated pathways included in the larger GRN (*Figure 5—figure supplement 1*). FZD6 is a transmembrane protein which functions as a receptor for WNT signalling proteins to, most commonly, activate the non-canonical planar cell polarity pathway. However, elements of the Wnt signalling pathway have been implicated to act antagonistically to chondrogenesis (*Corda and Sala, 2017*; *Snelling et al., 2016*). ITGA3 is a cell surface integrin which forms a heterodimer with ITGB1 to form α3β1 heterodimers through which chondrocyte–fibronectin ECM connections can be created (*LaPointe et al., 2013*; *Loeser, 2014*). Conditional deletion of *Itgb1* in cartilage impacts profoundly on skeletogenesis in mice (*Aszodi et al., 2003*). α3 integrins have been found to increase in OA cartilage, though a direct mechanism between ITGA3 regulating SOX9 is not clear. Regulation of SOX9 via CAV1 has been more studied, which has shown that the relationship between SOX9 and CAV1 is complex and requires further testing. CAV1 may be affecting SOX9 levels, via its activation of RHOA/ROCK1 signalling, which leads to phosphorylation of SOX9. RHOA/ROCK1 inhibition has been shown to both increase levels of SOX9 and chondrogenesis biomarkers in mouse ATDC5 cells, but also decrease levels of *SOX9* and chondrogenesis biomarkers in 3D-chondrocytes (*Peng et al., 2008*; *Xu et al., 2012*; *Woods and Beier, 2006*; *Woods et al., 2005*). It is likely that other mRNA targets of miR-199a/b-5p also contributed towards chondrogenesis regulation, such as the other genes identified as miR-199a/b-5p targets from the RNAseq analysis, for example, *MYH9*, *NECTIN2*. Based on this limitation, a 'blackbox' called OtherTargets was added to the kinetic model to represent all other anti-chondrogenic miR-199a/b-5p targets during chondrogenesis which were not explored in this study. A major limitation of the kinetic models is that we were not able to provide any multi-omic data – as no protein level, or phosphor-protein data were available. We generated a broader GRN (*Figure 5—figure supplement 1*), which showcases how the models we created could be enhanced with such data.

While the in silico model can serve as a resource for researchers interested in this system, there were certain genes such as *CAV1* and *FZD6* which proved difficult to model. At the time of building the models we assumed, miR-199a-5p inhibition would lead to a greater effect than miR-199b-5p inhibition due to miR-199a-5p being more abundant. However, we clearly see now this was a misconception and miR-199b-5p inhibition leads to a greater decrease in GAG levels and a greater increase in *FZD6*, *ITGA3,* and *CAV1* levels. This could be because miR-199b-5p increases by a greater magnitude than miR-199a-5p; therefore, the inhibition of miR-199b-5p has a bigger effect on chondrogenesis. Such time-series analysis would have been unavailable by only using differential expression analysis, and the *TimiRGeN* R package was highly useful in finding this microRNA during our reanalysis.

Our results validate miR-199a/b-5p interacting with *FZD6*, *ITGA3,* and *CAV1* and for miR-199a/b-5p to provide vital pro-chondrogenic regulatory effects, as observed previously for miR-140 and miR-455. Deletion of miR-140 in both humans and mice affects skeletal development (*Miyaki et al., 2010*; *Grigelioniene et al., 2019*). Further, from in vivo mouse models miR-140 and miR-455 were additionally shown to be pivotal in protecting from OA. Recently, intra-articular injection of a miR-199a-5p mimic has been shown to reduce cartilage damage in a rat post-traumatic OA model (*Huang et al., 2023*). Mouse genetic studies examining the loss of both –199a/b-5p, specifically in cartilage, are required to better understand the function of these miRNAs in skeletogenesis and chondrocyte development. Our results demonstrate early interest and provide a detailed kinetic model to aid researchers interested in this important topic.

## Conclusion

Our combined bioinformatic, laboratory, and systems biology methodology was a multi-faceted approach to explore miR-199a-5p and miR-199b-5p as pro-chondrogenic regulators. Based on our bioinformatic analysis, the three most significantly positively changing miR-199a/b-5p were predicted targets *FZD6*, *ITGA3,* and *CAV*. Laboratory experiments validated these as direct miR-199a/b-5p targets and confirmed that miR-199a/b-5p positively regulate chondrogenesis. However, the complex nature of miRNA function means there are likely multiple mRNA targets of miR-199a/b-5p which may work synergistically to modulate chondrogenesis. The GRN and kinetic models were created to capture the behaviours of the system which act as a useful resource for further experimental design.

## Materials and methods

### Data processing and differential expression analysis

mRNA and miRNA data were produced using Illumina and Exiqon microarray technologies, respectively (*Barter et al., 2015*). mRNA data was processed using the *lumi* R package, and the miRNA data was processed using the *affy* R package (*Du et al., 2008*; *Gautier et al., 2004*). *limma* was then used to perform pairwise differential expression (DE) analysis (*Ritchie et al., 2015*). Here, the zero timepoint was used as the common denominator for all DE analyses. The timepoint-based DE analyses were D1/D0, D3/D0, D6/D0, D10/D0, and D14/D0. This type of timepoint-based DE analysis was ideal for pairwise differential expression analysis approach, as explained by *Spies et al., 2019*. Genes which were significantly differentially expressed (BH adjusted p-values<0.05) in at least one of the DE analyses have their adjusted p-values and log2FC values were kept for analysis by the *TimiRGeN R* package. All data wrangling and processing took place in *R*.

### Analysis with the *TimiRGeN R* package

Dataframes containing mRNA and miRNA DE results were analysed using the combined mode of *TimiRGeN* (*Patel et al., 2021*). The threshold for timepoint-specific gene filtration was set as <0.05 and adjusted p-values were used for filtration. Timepoint-specific pathway enrichment used microarray probe IDs as the background dataset. Our data was from microarrays, so for more accurate overrepresentation analysis we required the microarray-specific gene lists to use as the background set of genes. GPL10558 (mRNA) and GPL11434 (miRNA) were downloaded from GEO, and the probes were annotated with entrez gene IDs using *BiomaRt* (*Durinck et al., 2009*; *Edgar et al., 2002*). Eight signalling pathways were enriched for at least three of the five timepoint-based DE analyses performed. miRNA–mRNA interactions were retained if they had a negative Pearson correlation of <–0.75 and if they identified two of the three target databases used by *TimiRGeN*.

### Human bone marrow MSC culture

Human bone marrow MSCs (from seven donors, 18–25 years of age, three females and four males) were isolated from human bone marrow mononuclear cells (Lonza Biosciences) and cultured and phenotype-tested as described previously (*Barter et al., 2015*). Experiments were performed using cells between passages 2 and 10, and all experiments were repeated with cells from a minimum of three donors.

### Chondrogenic differentiation

MSCs were resuspended in chondrogenic culture medium consisting high-glucose DMEM containing 100 µg/ml sodium pyruvate, 10 ng/ml TGFβ3, 100 nM dexamethasone, 1×ITS-1 premix, 40 µg/ml proline, and 25 µg/ml ascorbate-2-phosphate. $5 \times 10^5$ MSCs in 500 µl chondrogenic medium were pipetted into 15 ml falcon tubes or $5 \times 10^4$ MSCs in 150 µl chondrogenic medium were pipetted into a UV-sterilised V-bottom 96-well plate, and then centrifuged at $500 \times g$ for 5 min. Media were replaced every 2 or 3 days for up to 7days.

### Dimethylmethylene blue assay

Chondrogenic pellets and transwell discs were digested with papain (10 U/ml) at 60°C (*Murdoch et al., 2007*). The sulphated GAG content was measured by 1,9-dimethylmethylene blue binding (Sigma) using chondroitin sulphate (Sigma) as standard (*Farndale et al., 1982*).

### RNA and miRNA extraction and real-time reverse transcription PCR

MSC chondrogenic pellets were disrupted in Ambion Cells-to-cDNA II Cell Lysis buffer (for real-time RT-PCR) or mirVana miRNA Isolation Kit Phenol (for RNA-seq) (both Life Technologies) using a small disposable plastic pestle and an aliquot of Molecular Grinding Resin (G-Biosciences/Genotech). Total RNA was then extracted and converted to cDNA using MMLV reverse transcriptase (Invitrogen) and TaqMan real-time RT-PCR was performed, and gene expression levels were calculated as described previously (*Barter et al., 2010*). Primer sequences and assay details can be found in the supporting information materials. For single miRNA-specific analysis, RNA was reverse-transcribed with Applied Biosystems TaqMan MicroRNA Reverse Transcription Kit (Life Technologies) and real-time RT-PCR

performed with TaqMan MicroRNA assays (Life Technologies). All values are presented as the mean ±SEM of replicates in pooled experiments. For experiments with multiple MSC donors, statistical testing was performed using a matched paired two-tailed Student's t-test on log-transformed values to account for non-normal distribution. Primer details are in *Supplementary file 1e*.

## miRNA mimic/inhibitor transfection in MSC

For modulation of miR-199 levels in MSC, Dharmacon miRIDIAN mimics (C-300607) or miRIDIAN hairpin inhibitors (IH-300607) were transfected into 40–50% confluent MSC using Dharmafect 1 lipid reagent (Horizon Discovery) at 100 nM final concentration. Analysis was performed in comparison with Dharmacon miRIDIAN miRNA mimic nontargeting Control #2 (CN-002000-01) or Dharmacon miRIDIAN miRNA hairpin inhibitor nontargeting Control #2 (IN-002005-01). For all experiments, cells were subject to a single transfection prior to induction of MSC differentiation.

## Cloning and plasmid transfection in SW1353 cells

SW1353 cells were purchased from ATCC (https://www.atcc.org/products/htb-94). Cell identity is authenticated by ATCC by STR profiling. Cells are routinely checked for mycoplasma status (MycoAlert Mycoplasma Detection Kit, Lonza Biosciences). Full-length miRNA target 3′UTRs were amplified from human genomic DNA using PCR primers (*Supplementary file 1e*) to enable Clontech In-Fusion HD cloning (Takara Bio Europe, Saint-Germain-en-Laye, France) into the pmirGLO Dual-Luciferase miRNA Target Expression Vector (Promega, Southampton, UK) following the manufacturer's instructions. Mutation of the miRNA seed-binding sites was performed using the QuikChange II Site-Directed Mutagenesis Kit (Agilent Technologies) (*Supplementary file 1e*). All vectors were sequence verified. SW1353 chondrosarcoma cells were plated overnight in 96-well plates at 50% confluence ($1.5 \times 10^4$ cells/cm$^2$). Cells were first transfected with 3′UTR luciferase constructs (10 ng) using FuGENE HD transfection reagent (Promega) for 4 hr, then transfected with Dharmacon *miR-199a-5p* mimic (50 nM) or miRNA mimic nontargeting Control #2 using Dharmafect 1. After 24 hr of transfection, the SW1353 cells were washed and lysed using Reporter Lysis Buffer (Promega) and firefly and renilla luciferase levels determined using the Promega Dual-Luciferase Reporter Assay System and a GloMax 96 Microplate Luminometer (Promega).

## RNAseq

RNA isolated as described above was quality assessed with the Agilent Technology 4200 TapeStation system using an RNA screentape assay (Agilent). cDNA libraries were generated using the Illumina TruSeq Stranded mRNA protocol, combinatorial dual index adapters were used to multiplex/pool libraries. Single-read sequencing, 76 cycles (75 + 1 cycle for index sequences), on an Illumina NextSeq500 instrument using a high-output 75 cycle kit. *Kallisto* was used for alignment free RNAseq processing (*Bray et al., 2016*). *Tximport* was used to import the RNAseq data into *R* for further processing (*Soneson et al., 2015*).

## RNAseq DE and miRNA target identification

Time-matched MSC miRNA inhibition (miR-199a-5p and miR-199b-5p) and MSC controls were contrasted by DE using *DESeq2* (*Love et al., 2014*). *miRNAtap* was used to score and identify all potential mRNA targets of both miR-199a-5p and miR-199b-5p (*Pajak and Simpson, 2021*). Using several target databases, potential mRNA targets were scored and ranked. Low-level scoring miRNA targets (50 or below) and negatively changing genes from the time-series dataset were removed, leaving 21 genes that were significantly (adjusted p-values<0.05) upregulated during day 0 or day 1 following miR-199a-5p and miR-199b-5p inhibition.

## Chi-square tests

To determine if the number of overlapping genes found from differential expression analysis of 199a/b inhibition at days 0 and 1 were significant, chi-square tests were performed. Observed numbers were used to determine the estimated numbers. Expected number of differentially expressed overlapping/non-overlapping genes = (Row Total * Column Total/Grand Total). Chi-square p-value calculation is performed with the following formula: (Observed – Expected)^2/Expected.

## GRN development

For the enhanced kinetic model, we used *CellDesigner*, SMBL-style GRNs are created to represent the biological processed of interest (*Funahashi et al., 2008*). A literature search provided information of GRN topology. Once the whole GRN was created (*Figure 5C*), a more advanced GRN was created to hypothesise further pathways and important players in the system which could not be modelled as we lacked the experimental data to do so (*Figure 5—figure supplement 1*).

## Kinetic modelling

Base data (microarray experiments) and the validation data (*Figures 2 and 4*) from qPCR data were converted to numbers compatible to the calibration data from the microarrays using the following formula: $KD/C * M$, $KD$ being the mean miR-199a-5p or miR-199b-5p value, $C$ being the mean control value, and M being the mean microarray value. The initial conditions (zero timepoint) for the calibration and validation datasets were assumed to be the same, so the model can have a single initial condition for each validated species. Thus, $KD/C = 1$ was fixed for each zero timepoint, for each species where validation data was available. For GAG levels, the control level at day 7 was treated as 100%, and the change in GAG expression during miR-199b-5p inhibition was measured in contrast to the control to calculate the percentage change. Species selected from the modelled GRN were modelled using *COPASI* (*Hoops et al., 2006*). Calibration was performed using parameter estimation via the Particle swarm algorithm. Inhibition of miR-199a-5p and miR-199b-5p was simulated using events, where the miRNA in question was reduced by 90–95%, until day 7 in the initial model and day 4.5 in the enhanced model. Parameters were altered using sliders to make the model perform miR-199a-5p or miR-199b-5p inhibition behaviour. MSE was calculated between actual and simulated data where possible, $1/n \cdot \sum_{i=1}^{n} (yi - \hat{y}i)$ (*Akkiraju and Nohe, 2015*). Model formulas and parameters are in the supplementary materials. Data from *COPASI* was imported into *R* for plotting. All model parameters and equations have been recorded in the supplementary materials, and the model has been uploaded onto the Biomodels public repository (*Malik-Sheriff et al., 2020*).

## Acknowledgements

KP, IMC, MJB, and DY were supported by the Dunhill Medical Trust (R476/0516). DPS was supported by Novo Nordisk Fonden Challenge Programme: Harnessing the Power of Big Data to Address the Societal Challenge of Aging (NNF17OC0027812). CP and DY were supported by the MRC and Versus Arthritis as part of the Medical Research Council Versus Arthritis Centre for Integrated Research into Musculoskeletal Ageing (CIMA) (MR/P020941/1 and MR/R502182/1). JS received funding from Versus Arthritis (22043).

## Additional information

### Funding

| Funder | Grant reference number | Author |
| --- | --- | --- |
| Dunhill Medical Trust | R476/0516 | Krutik Patel<br>Matt Barter<br>David Young<br>Daryl P Shanley |
| Novo Nordisk Fonden | NNF17OC0027812 | Daryl P Shanley |
| Versus Arthritis | MR/P020941/1 | Carole Proctor<br>David Young |
| Versus Arthritis | MR/R502182/1 | Carole Proctor<br>David Young |
| Versus Arthritis | 22043 | Jamie Soul |

| Funder | Grant reference number | Author |
|---|---|---|

The funders had no role in study design, data collection and interpretation, or the decision to submit the work for publication.

## Author contributions

Krutik Patel, Conceptualization, Resources, Data curation, Software, Formal analysis, Validation, Investigation, Visualization, Methodology, Writing – original draft, Writing – review and editing; Matt Barter, Conceptualization, Data curation, Validation, Investigation, Methodology, Writing – review and editing; Jamie Soul, Data curation, Methodology, Writing – review and editing; Peter Clark, Data curation; Carole Proctor, Supervision; Ian Clark, Supervision, Funding acquisition; David Young, Conceptualization, Supervision, Project administration, Writing – review and editing; Daryl P Shanley, Conceptualization, Supervision, Funding acquisition, Project administration

## Author ORCIDs

Krutik Patel ⓘ https://orcid.org/0000-0001-6806-8675
Ian Clark ⓘ https://orcid.org/0000-0003-2619-0896
David Young ⓘ http://orcid.org/0000-0002-7078-6745

Reviewer #1 (Public review): https://doi.org/10.7554/eLife.89701.4.sa1
Author response https://doi.org/10.7554/eLife.89701.4.sa2

# Additional files

## Supplementary files

• Supplementary file 1. A collection of seven tables which contain necessary information for this research. (**a**) Frequency of significant pathways enriched across the chondrogenesis time course. (**b**) 100 most significantly differentially expressed genes from differential expression analysis which contested day 0 and day 1 chondrogenesis/non-chondrogenesis samples. (**c**) From the time-course chondrogenesis dataset, we view the log2fc values from 21 predicted 199a/b targets. (**d**) From two independent chondrogenesis datasets, we view the log2fc values from 21 predicted 199a/b targets. (**e**) Primers and probes used for the knockdown experiments. (**f**) Model details for the initial model, including species and ODEs. (**g**) Model details for the enhanced model, included species, and ODEs.

• MDAR checklist

## Data availability

All scripts will be found in the following GitHub repository: https://github.com/Krutik6/MIR199ab5p-Chondrogenesis-Modelling-Paper/ (copy archived at *Patel, 2024*). The enhanced kinetic model can be found in the following biomodels repository MODEL2305010001. RNAseq data generated in this project can be found in GSE274379.

The following datasets were generated:

| Author(s) | Year | Dataset title | Dataset URL | Database and Identifier |
|---|---|---|---|---|
| Young D, Barter M, Soul J, Shanley D, Patel K | 2024 | Systems analysis of miR-199a/b-5p and multiple miR-199a/b-5p targets during chondrogenesis | https://www.ncbi.nlm.nih.gov/geo/query/acc.cgi?acc=GSE274379 | NCBI Gene Expression Omnibus, GSE274379 |
| Patel K | 2024 | Patel2023 - miRNA-199a/b-5p regionation of chondrogenesis - enchanced | https://www.ebi.ac.uk/biomodels/MODEL2305010001 | EBI BioModels, MODEL2305010001 |

The following previously published datasets were used:

| Author(s) | Year | Dataset title | Dataset URL | Database and Identifier |
|---|---|---|---|---|
| Huang AH, Stein A, Mauck RL | 2010 | Comparison of bovine undifferentiated MSCs, chondrogenic MSCs and chondrocytes in 3D culture | https://www.ncbi.nlm.nih.gov/geo/query/acc.cgi?acc=GSE18394 | NCBI Gene Expression Omnibus, GSE18394 |
| Huynh NPT, Zhang B, Guilak F | 2019 | High-Depth Transcriptomic Profiling Reveals the Temporal Gene Signature of Mesenchymal Stem Cells During Chondrogenesis | https://www.ncbi.nlm.nih.gov/geo/query/acc.cgi?acc=GSE109503 | NCBI Gene Expression Omnibus, GSE109503 |

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
