## [Editor Report · eLife assessment]

This study provides **valuable** insight into the role of miR-199a/b-5p in cartilage formation. The evidence supporting the significance of the identified miRNA and its target mRNA transcripts is **convincing**. This article will likely primarily benefit scientists focused on diseases related to this biological process, such as osteoarthritis. Furthermore, researchers with a broader interest in miRNAs may find the computational model to identify novel RNA–RNA interactions particularly helpful.

---

## [Referee Report · Reviewer #1 (Public review)]

The comments below are from my review of the first submission of this article. I would now like to thank the authors for their hard work in responding to my comments. I am happy with the changes they have made, in particular the inclusion of further experimental evidence in Figures 2 and 4. I have no further comments to make.

In 'Systems analysis of miR-199a/b-5p and multiple miR-199a/b-5p targets during chondrogenesis', Patel et al. present a variety of analyses using different methodologies to investigate the importance of two miRNAs in regulating gene expression in a cellular model of cartilage development. They first re-analysed existing data to identify these miRNAs as one of the most dynamic across a chondrogenesis development timecourse. Next, they manipulated the expression of these miRNAs and showed that this affected the expression of various marker genes as expected. An RNA-seq experiment on these manipulations identified putative mRNA targets of the miRNAs which were also supported by bioinformatics predictions. These top hits were validated experimentally and, finally, a kinetic model was developed to demonstrate the relationship between the miRNAs and mRNAs studied throughout the paper.

I am convinced that the novel relationships reported here between miR-199a/b-5p and target genes FZD6, ITGA3 and CAV1 are likely to be genuine. It is important for researchers working on this system and related diseases to know all the miRNA/mRNA relationships but, as the authors have already published work studying the most dynamic miRNA (miR-140-5p) in this biological system I was not convinced that this study of the second miRNA in their list provided a conceptual advance on their previous work.

I was also concerned with the lack of reporting of details of the manipulation experiments. The authors state that they have over-expressed miR-199a-5p (Figure 2A) and knocked down miR-199b-5p (Figure 2B) but they should have reported their proof that these experiments had worked as predicted, e.g. showing the qRT-PCR change in miRNA expression. Similarly, I was concerned that one miRNA was over-expressed while the other was knocked down - why did the authors not attempt to manipulate both miRNAs in both directions? Were they unable to achieve a significant change in miRNA expression or did these experiments not confirm the results reported in the manuscript?

I had a number of issues with the way in which some of the data is presented. Table 1 only reported whether a specific pathway was significant or not for a given differential expression analysis but this concealed the extent of this enrichment or the level of statistical significance reported. Could it be redrawn to more similarly match the format of Figure 3A? The various shades of grey in Figure 2 and Figure 4 made it impossible to discriminate between treatments and therefore identify whether these data supported the conclusions made in the text. It also appeared that the same results were reported in Figure 3B and 3C and, indeed, Figure 3B was not referred to in the main text. Perhaps this figure could be made more concise by removing one of these two sets of panels?

Overall, while I think that this is an interesting and valuable paper, I think its findings are relatively limited to those interested in the role of miRNAs in this specific biomedical context.

---

## [Author Response]

The following is the authors’ response to the previous reviews.

**Public Review:**

**Reviewer #1 (Public Review):**
In 'Systems analysis of miR-199a/b-5p and multiple miR-199a/b-5p targets during chondrogenesis', Patel et al. present a variety of analyses using different methodologies to investigate the importance of two miRNAs in regulating gene expression in a cellular model of cartilage development. They first re-analysed existing data to identify these miRNAs as one of the most dynamic across a chondrogenesis development time course. Next, they manipulated the expression of these miRNAs and showed that this affected the expression of various marker genes as expected. An RNA-seq experiment on these manipulations identified putative mRNA targets of the miRNAs which were also supported by bioinformatics predictions. These top hits were validated experimentally and, finally, a kinetic model was developed to demonstrate the relationship between the miRNAs and mRNAs studied throughout the paper.I am convinced that the novel relationships reported here between miR-199a/b-5p and target genes FZD6, ITGA3, and CAV1 are likely to be genuine. It is important for researchers working on this system and related diseases to know all the miRNA/mRNA relationships but, as the authors have already published work studying the most dynamic miRNA (miR-140-5p) in this biological system I was not convinced that this study of the second miRNA in their list provided a conceptual advance on their previous work.

We believe this study is an enhancement on our previous work for two reasons, which have been alluded to in new text within the introduction. Firstly, our previous work used experimental and bioinformatic analysis to identify microRNAs with significant regulatory roles during chondrogenesis. This new manuscript additionally uses a systems biology approaches to identify novel miRNA-mRNA interactions and capture these within an in silico model. Secondly, this work was initiated by the analysis of our previously generated data – using a novel tool we developed for this type of data (Bioconductor - TimiRGeN).

I was also concerned with the lack of reporting of details of the manipulation experiments. The authors state that they have over-expressed miR-199a-5p (Figure 2A) and knocked down miR-199b-5p (Figure 2B) but they should have reported their proof that these experiments had worked as predicted, e.g. showing the qRT-PCR change in miRNA expression. Similarly, I was concerned that one miRNA was over-expressed while the other was knocked down - why did the authors not attempt to manipulate both miRNAs in both directions? Were they unable to achieve a significant change in miRNA expression or did these experiments not confirm the results reported in the manuscript?

We agree with the reviewer that some additional data were needed to demonstrate the effective regulation of miR-199-5p. Hence, Supplementary Figure 1 is now included which provides validation of the effects of miR-199a-5p overexpression

(Supplementary Figure 1A) and inhibition of miR-199a/b-5p (Supplementary Figure 1B). Within the main manuscript, Figure 2B has been amended to include the consequences of inhibition of miR-199a-5p, with 2C showing the consequences of miR-199b-5p inhibition. Further, we include new data with regards to miR-199a/b-5p inhibition on CAV1 (Figure 4A).

I had a number of issues with the way in which some of the data was presented. Table 1 only reported whether a specific pathway was significant or not for a given differential expression analysis but this concealed the extent of this enrichment or the level of statistical significance reported. Could it be redrawn to more similarly match the format of Figure 3A? The various shades of grey in Figure 2 and Figure 4 made it impossible to discriminate between treatments and therefore identify whether these data supported the conclusions made in the text. It also appeared that the same results were reported in Figure 3B and 3C and, indeed, Figure 3B was not referred to in the main text. Perhaps this figure could be made more concise by removing one of these two sets of panels.

We agree with all points made here and have amended these within the manuscript. Figure 1A is now pathway enrichment plots from the TimiRGeN R Bioconductor package, and the table which previously showed the pathways enriched at each time point is now in the supplementary materials (supp. Table 1). Figure 2 and 4 now have color instead of shades of grey. Figure 3C has now been moved to supplementary materials (Supplementary Figure 2) and is referenced in the text.

Overall, while I think that this is an interesting and valuable paper, I think its findings are relatively limited to those interested in the role of miRNAs in this specific biomedical context.
**Reviewer #2 (Public Review):**
Summary:This study represents an ambitious endeavor to comprehensively analyze the role of miR199a/b-5p and its networks in cartilage formation. By conducting experiments that go beyond in vitro MSC differentiation models, more robust conclusions can be achieved.Strengths:This research investigates the role of miR-199a/b-5p during chondrogenesis using bioinformatics and in vitro experimental systems. The significance of miRNAs in chondrogenesis and OA is crucial, warranting further research, and this study contributes novel insights.Weaknesses:While miR-140 and miR-455 are used as controls, these miRNAs have been demonstrated to be more relevant to Cartilage Homeostasis than chondrogenesis itself. Their deficiency has been genetically proven to induce Osteoarthritis in mice. Therefore, the results of this study should be considered in comparison with these existing findings.

We agree with the reviewers comments. miR-455-null mice develop normally but miR-140-null (or mutated) mice and humans do have skeletal abnormalities (e.g. Nat Med. 2019 Apr;25(4):583-590. doi: 10.1038/s41591-019-0353-2), indicating a role in chondrogenesis. We have made an addition in the description to point towards the need to assess the roles miR-199a/b-5p may play during skeletogenesis and OA. We anticipate miR-199a/b-5p to be relevant in OA and have ongoing additional work for this – but this beyond the scope of this manuscript.

**Recommendations For The Authors:**

**Reviewer #1 (Recommendations For The Authors):**
Beyond the issues raised in the public review, I had a few minor recommendations that are largely designed to help improve the understanding of the manuscript as it is currently written.(1) Please provide the statistical tests used to obtain p-values in the Figure 2 and 4 legends.

We have now added statistical test information to the figure legends of figures 2 and 4.

(2) It is stated on p. 9 that both miRNAs may share a functional repertoire because 25 and 341 genes are interested between their inhibition experiments. Please provide statistical support that this overlap is an enrichment over the null background in this experiment. Total DE genes – chi squared. Expected / Observed.

A chi-squared test is now presented in the manuscript which shows that the number of significant genes which were found in common between miR-199a-5p knockdown and miR-199b-5p knockdown were significantly more than expected for day 0 or day 1 of the experiments.

(3) The final sentence on p. 12 (beginning 'Size of the points reflect...') seemed out of place - is it part of a legend?

Thank you for pointing out this mistake - it was part of figure 3C and now is in the supplementary materials.

(4) A sentence on p. 14 reads that 'FZD6 and ITGA3 levels increased significantly' but this should read decreased, rather than increased. Quite an important typo!

Thank you for pointing this error out. It has been corrected.

(5) Theoretical transcripts are mentioned in the legend of Figure 5A but these were not present in the figure. Please include these or remove them from the legend.

This error has been removed form Figure 5A.

(6) On p 20, the references 22 and 27 should I think be moved to earlier in the sentence (after 'miR-199a-5p-FZD6 has been predicted previously'). Currently, it reads as if these references support your luciferase assays which you claim are the first evidence for this target relationship.

We agree with this change and have corrected the manuscript.

(7) The reference to Figure 5D on p. 20 should be a reference to Figure 5C.

Thank you for pointing this error out – this has been corrected.

**Reviewer #2 (Recommendations For The Authors):**
(1) The paper is based on the importance of miR-140 and miR-455 as miRNAs in chondrogenesis, citing only Barter, M. J. et al. Stem Cells 33, (2015). Considering the scope and results of this study, this citation is insufficient.

We agree with this reviewers comments. For many year miR-140 and miR-455 have been experimented on and their importance in OA research has become apparent. We included additional references within the introduction to address this.

(2) Analyzing chondrogenesis solely through differentiation experiments from MSCs is inadequate. It is essential to perform experiments involving the network within normal cartilage tissue and/or the generation of knockout mice to understand the precise role of miR199a/b-5p in chondrogenesis.

We have added an additional paragraph in the discussion to state this, and do believe it is highly important that miR-199a/b-5p be tested in OA samples – however this would be beyond the intended scope of this article.

(3) In light of the above points, it is imperative to investigate the role of miR-199a/b-5p beyond the in vitro differentiation model from MSCs, encompassing mouse OA models or human disease samples.

In tangent with the previous address, we agree with the pretense and believe additional experiments should be performed to gain more insight to the mechanism of how miR-199a/b-5p regulate OA. But development of a new mouse line to investigate this is not in the scope of this manuscript.